# Effect of Upper Limb Motor Rehabilitation on Cognition in Parkinson’s Disease: An Observational Study

**DOI:** 10.3390/brainsci12121684

**Published:** 2022-12-08

**Authors:** Valentina Varalta, Elisa Evangelista, Anna Righetti, Giovanni Morone, Stefano Tamburin, Alessandro Picelli, Cristina Fonte, Michele Tinazzi, Ilaria Antonella Di Vico, Andreas Waldner, Mirko Filippetti, Nicola Smania

**Affiliations:** 1Neuromotor and Cognitive Rehabilitation Research Center, Section of Physical and Rehabilitation Medicine, Department of Neurosciences, Biomedicine and Movement Sciences, University of Verona, 37134 Verona, Italy; 2Neurorehabilitation Unit, University Hospital of Verona, 37134 Verona, Italy; 3Department of Life, Health and Environmental Sciences, University of L’Aquila, 67100 L’Aquila, Italy; 4San Raffaele Sulmona Institute, 67039 Sulmona, Italy; 5Section of Neurology, Department of Neurosciences, Biomedicine, and Movement Sciences, University of Verona, 37134 Verona, Italy; 6Canadian Advances in Neuro-Orthopedics for Spasticity Congress (CANOSC), Kingston, ON K7K 1Z6, Canada; 7Neurology Unit, USD Parkinson e Disturbi del Movimento, University Hospital of Verona, 37134 Verona, Italy; 8Department of Neurological Rehabilitation, Private Hospital “Villa Melitta”, Via Col di Lana 6, 39100 Bolzano, Italy

**Keywords:** neuropsychological deficits, cognitive motor interference, motor treatment, movement disorders, vibratory stimulation

## Abstract

Parkinson’s disease is characterized by motor and cognitive deficits that usually have an impact on quality of life and independence. To reduce impairment, various rehabilitation programs have been proposed, but their effects on both cognitive and motor aspects have not been systematically investigated. Furthermore, most intervention is focused on lower limb treatment rather than upper limbs. In the present study, we investigated the effect of 3-week upper limb vibratory stimulation training on cognitive functioning in 20 individuals with Parkinson’s disease. We analyzed cognitive (Montreal Cognitive Assessment, Trial Making Test, Digit Symbol, Digit Span Forward and Backward and Alertness) and motor performance (Unified Parkinson’s Disease Rating Scale—part III; Disability of the Arm, Shoulder and Hand Questionnaire) before treatment, at the end of treatment and one month post treatment. After rehabilitation, a statistically significant improvement was observed in terms of global cognitive status, attention, global motor functioning and disability. The results suggest an impact of upper limb motor rehabilitation on cognition in Parkinson’s disease. Future studies on neuromotor interventions should investigate their effects on cognitive functioning to improve understanding of cognitive motor interaction in Parkinson’s disease.

## 1. Introduction

Parkinson’s disease (PD) is a neurological degenerative disease that leads to motor and non-motor impairments. Motor aspects of PD include problems with gait, postural control and tremors [1,2]. Furthermore, some people with PD (pwPD) present with cognitive difficulties in executive function, memory and attention [3,4,5], affecting approximately 25% of newly diagnosed patients [5,6]

Pharmacological interventions and deep brain stimulation treatment are not effective in slowing or treating the progression of cognitive decline in pwPD [7,8]. Physical exercise is a valid complementary therapy option not only to alleviate motor symptoms [9,10] but also to exert a beneficial impact on cognitive impairment, especially in the early stage of disease in healthy older adults [11,12,13]. Clinical and preclinical studies on animal and human models have highlighted multiple mechanisms by which physical activity may improve brain function, with a consequent positive impact on both motor and cognitive aspects in pwPD and healthy older rats [14,15,16]. Furthermore, some studies have suggested that the neural networks underlying such cognitive processes may also play an important role in motor manifestations and vice versa, highlighting the role of cognition in determining/influencing some motor symptoms [17,18]. A recent systematic review of randomized controlled trials [13] reported positive effects of physical exercise on cognitive functions in pwPD with aerobic exercises [19], robotic training [14,20] and dual-task training [21]. Specifically, global cognitive function [19,21] and specific aspects of executive and spatial functions, such as processing speed, sustained attention and cognitive flexibility, [14,20] were observed at the end of training. Physical therapy programs seem to support the stabilization of an individual’s cognitive performance over time [22]. For example, cognitive improvement observed immediately after treatment was shown not to be maintained at six-month follow-up. However, the neuropsychological functional level of the experimental group returned to baseline, in contrast to the control group, which presented with a progression of cognitive decline [22].

Many studies have demonstrated the influence of motor training on cognition in people with neurological diseases [23,24,25]. In pwPD, cognitive functioning has been investigated after various kinds of aerobic, walking or dance training [26,27]. The data indicate that the involvement of the lower limbs has an influence specifically on executive function, verbal memory, language and attention [26,28].

Regarding the effect of upper limb motor therapy on cognitive functioning, some studies have demonstrated the influence of motor rehabilitation on specific cognitive abilities of people with stroke, such as short-term memory, working memory and attention [23,29]. Daily upper extremity training improves short-term memory, attention and executive functions [29]. To the best of our knowledge, no studies have been published to date that have investigated the possible effects of upper limb motor therapy on cognition in pwPD.

Upper limb deficits in such individuals are usually treated by fine-motor hand exercise programs, practice-based programs, visual and auditory cueing [30], resistance training [31] and rhythmic auditory stimulation [32].

Another rehabilitation approach in pwPD is vibration stimulation. However, this method is typically used to improve gait and balance [33]. The underlying mechanism is the induction of a series of vibratory stimuli that favor a central integration and elaboration of proprioceptive stimuli and improved motor and postural control [33,34]. A meta-analysis reported on improvements in gait and balance after whole-body vibration conducted with different oscillation parameters. The authors suggested that the results could be explained by the activation of vibratory-induced muscular receptors and the related motor response, which could contribute to a major central sensitization and improved sensory elaboration, producing improvements in the adaptation of postural adjustments [33]. This mechanism could have an effect on neuropsychological aspects; however, to the best of our knowledge, there are no data available with respect to cognitive functioning. We therefore hypothesize a possible influence of vibratory treatment on cognitive functioning, as well as on motor aspects.

The aim of the present study was to investigate whether upper limb vibratory stimulation training has an influence on cognitive functioning in pwPD.

## 2. Materials and Methods

### 2.1. Study Design

This uncontrolled before–after clinical trial was carried out at the Neurorehabilitation Unit of the University Hospital of Verona from January 2022 to July 2022. The study was approved by the CESC (Comitato Etico per la Sperimentazione Clinica delle Province di Verona e Rovigo; Approval code: 3670CESC). All participants were informed about the study’s procedures and provided written informed consent before taking part in the assessment. A consecutive sample was used. The participants enrolled in this study are a subgroup involved in the registered randomized control trial on Open Science Framework—OSF (https://doi.org/10.17605/OSF.IO/CH8MB, accessed on 1 December 2022). The study protocol was performed in accordance with the Declaration of Helsinki.

### 2.2. Subjects

Individuals with a diagnosis of PD according to United Kingdom Parkinson’s Disease Society Brain Bank Criteria [35] were recruited. Participants presenting with disease stages 1 to 3 according to the Hoen and Yahr classification (determined in the “on” phase) [36] or absence of dementia according to the Montreal Cognitive Assessment (MoCA > 15.50) [37] were included. Exclusion criteria were other neurological disorders or orthopedic conditions involving upper limbs, concomitant neuromotor or cognitive rehabilitation treatment during the period of management for the study or carried out in the 2 preceding months, changes in drug therapy during the study period, psychiatric diseases, untreated depression or anxiety symptoms, alcohol or drug abuse and the presence of uncorrected visual or auditory disturbances that may limit the administration of the test and/or treatment.

### 2.3. Intervention and Procedures

Participants underwent a 3-week one-to-one treatment three times a week for a total of nine sessions. Each session lasted approximately 45 min and was led by a physiotherapist specialized in neuromotor rehabilitation. Treatment consisted of an upper limb exercise protocol using a device that produces various vibration frequencies (Armshake^®^ device, product by Move it GmbH; www.move-it-med.com, accessed on 1 December 2022) [38]. Vibration was transmitted to the entire upper limb of the subject and may have varied during the treatment (range: 2 to 20 Hz). The training session involved exercises performed with one or both upper limbs holding the device. Details about the exercises are provided in the Appendix A. During the activities, participants were encouraged to concentrate and focus attention on the position and movement of their upper limbs. In addition, during exercises performed in the standing position, participants were encouraged to maintain a postural stance as erect as possible in an effort to work on the coordination between the upper limbs and balance maintenance. Individuals who used assistive ambulation devices performed exercises mostly in the seated position.

The intervention was adapted to each individual according to his/her level of motor and cognitive functionality. Adaptations implemented in each session in terms of exercises difficulty (i.e., number of repetitions, duration of the exercises, rest time between exercised, seated/standing position, etc.). See the attached form for more details (Appendix A).

### 2.4. Data Collection, Assessment Procedures and Blinding

In addition to the motor assessment, each individual underwent 3 cognitive evaluations before (T0), immediately after (T1) and 30 days after the physiotherapy treatment (T2).

Cognitive measures were administered by a neuropsychologist (E.E.).

In the present study, we were specifically interested in the influence of motor treatment on cognitive functioning. In terms of motor outcome measures, we reported only data related to global motor functioning and disability of the upper limbs collected by a rehabilitation physician.

Outcome measure assessors were not involved in treatment and were blinded with respect to the timing of the assessment (i.e., T0, T1 or T2). The pwPD subjects were involved in a study with a repeated measure analysis of primary and secondary cognitive outcomes.

The participants were tested in the “on” state.

### 2.5. Outcome Measures

#### 2.5.1. Primary Outcome Measure

The primary outcome measure was the Trail Making Test, Part B (TMT B), which investigates attention capacity, the ability to switch attention between two rules or tasks and cognitive flexibility. The time taken to complete the trails was recorded (longer = worse performance) [39].

#### 2.5.2. Secondary Outcome Measures

Additional cognitive, motor and disability outcomes were recorded.

To screen the cognitive status of pwPD, we used the Italian version of the MoCA (ICC = 0.79; [40]) [37], which consists of 12 subtasks exploring the following cognitive domains: (1) memory, (2) visuospatial abilities, (3) executive functions, (4) attention, concentration and working memory, (5) language, and (6) temporal and spatial orientation. The MoCA total score ranges from 0 (worst performance) to 30 (best performance). To reduce possible learning effects between consecutive assessments, we used different versions of the MoCA (7.1, 7.2 and 7.3) [41].

Visual selective attention and sequencing skills were evaluated with the TMT (ICC part A = 0.745, part B = 0.849 [39]). The time taken to complete the trails was recorded (longer = worse performance) [39]. Digit Symbol (DSym) (ICC = 0.91), a subtest of the Wechsler Adult Intelligence Scale, was used to measure psychomotor speed, visual–motor coordination, attention and concentration. Within a specified time limit, the participants pressed a key to copy symbols that were paired with numbers (higher scores indicate better performance) [42]. Working memory was assessed with the Digit Span Backward (DSB) test [43]. Individuals were asked to repeat a list of single-digit numbers in the correct reverse order immediately after presentation. The maximum score is 8 (best performance). Reaction time was tested by under two conditions using Alertness, a subtest of the computerized Test of Attentional Performance 2.3.1 (TAP) (ICC = 0.90) [44]. The first condition involves simple reaction time measurements, measuring intrinsic alertness; a cross appears on the monitor at randomly varying intervals, to which the subject is instructed to respond as quickly as possible by pressing a key (Alertness without advertisement, ALL_SA). Under a second condition, reaction time is measured in response to a critical stimulus preceded by a cue stimulus presented as warning tone (Alertness with advertisement, ALL_CA).

Global motor functioning was assessed using the Unified Parkinson’s Disease Rating Scale—part III (MDS-UPDRS—part III) [36], which consists of 34 scores (each rated on a scale from 0 to 4 points) based on 18 questions about tremors, slowness (bradykinesia), stiffness (rigidity) and balance and was used to assess movement capacity. The total score is the sum of all items, ranging from 0 (best performance) to 136 (worst performance). The upper limb disability experiences of participant were investigated using the Disability of the Arm, Shoulder and Hand Questionnaire (DASH) (ICC = 0.536; [45]) [46]. This questionnaire lists a series of activities associated with daily living; respondents are asked to indicate their difficulty in carrying out a given activity on a scale from 1 (no difficulty) to 5 (I didn’t succeed). The total score ranges from 0 to 150 (higher score = higher level of felt disability).

### 2.6. Statistical Analysis

Data were analyzed using IBM SPSS version 26.0 software for Macintosh (IBM Corp, Armonk, NY, USA). Normal distribution of data was determined using Kolmogorov–Smirnov (KS) and Shapiro–Wilk (SW) tests. Normal variables were analyzed with one-way ANOVA for repeated measures within-individual factor “time” with 3 levels (pretreatment, post treatment, and follow-up). The alpha level for significance was set at *p* < 0.05. Post hoc comparisons were corrected using the least significant difference (LSD) method. Non-normally distributed variables were analyzed using the Wilcoxon signed-rank test. Results are presented as means and standard deviations. Power analysis was carried out with G*Power 3.1.9.4 software (Faul, F., Erdfelder, E., Lang, A.G., & Buchner, A. 2007; Heinrich-Heine-Universität Düsseldorf, Germany), with one group measured across three observations, an alpha of 0.05, a power of 0.80 and a small treatment effect of 0.01. A total of 16 observations were needed to detect a significant treatment effect. Therefore, a total sample size of 18 participants was suitable, taking into account a possible 15% rate of dropout.

## 3. Results

A total of 20 individuals (18 men, 2 women; age: 70.65 ± 8.5 years; education: 8.8 ± 3.42 years; H&Y classification 2.05 ± 0.88) presenting with idiopathic PD (disease duration: 5 ± 4.74 years; 6 presented with depression) were recruited among 36 individuals referred to the Neurorehabilitation Unit of University Hospital of Verona, Italy, between January 2022 and July 2022. Two individuals needed assistive devices (walker-rollator), and all participants were on dopaminergic medication. The pwPD were allocated for upper limb motor training. No dropout was observed, and no adverse events occurred during the trial. Individuals completed the training program and post-treatment evaluation, and 18 participants completed follow-up assessment. A flow diagram of the study is shown in Figure 1.

### 3.1. Baseline

MoCA, TMT B, DSym and ALL_CA were normally distributed (KS and SW tests, *p* > 0.05) and were analyzed by parametric tests.

### 3.2. Primary Outcome

One-way ANOVA revealed a principal significant effect of “time” for TMT B (F(2.32) = 3.988; *p* = 0.045; η = 0.2). Post hoc comparisons revealed significantly higher scores post treatment (mean score: 153.95 ± 74.96; *p* = 0.011) compared to pretreatment (mean score: 179.63 ± 81.19) and no change from baseline to follow-up (mean score: 208.76 ± 93.57; *p* = 0.389).

### 3.3. Secondary Outcomes

Within-group comparison showed significant changes in pretreatment versus post-treatment scores for the MDS-UPDRS—part III (*p* = 0.005; z = −2.827), DASH (*p* = 0; z = −3.298), TMT A (*p* = 0.011; z = −2.557), ALL_SA (*p* = 0.033; z = −2.134) and ALL_CA (*p* = 0.009; z = −2.628). Scores also significantly changed between pretreatment and the follow-up phase for DASH (*p* = 0.001; z = −3.298), TMT A (*p* = 0.015; z = −2.439), ALL_SA (*p* = 0.049; z = −1.965) and MDS-UPDRS—part III (*p* = 0.005; z = −2.204).

The one-way ANOVA revealed a significant main effect of time, with significant differences observed in DSym scores between pre- and post-treatment phases (F(2.32) = 6.758; *p* = 0.004). In particular, DSym scores were higher post treatment (M = 23.41; SD = 7.82, *p* = 0.003) compared to pretreatment (M = 20.94; SD = 8.36) and were higher in the follow-up phase (M = 23.71; SD = 9.47, *p* = 0.007) compared to pretreatment.

The other secondary outcome showed no significant differences. Table 1 and Table 2 present the group data and results of the within-group analysis.

## 4. Discussion

In the present study, we aimed to investigate the effects of upper limb vibratory stimulation training on cognitive functioning in pwPD.

The results indicate (1) an improvement in some specific cognitive abilities (i.e., executive functions and attention) and (2) motor abilities (i.e., global motor functioning and reduction in the disability of the upper limbs), whereas (3) no effect was observed on global cognitive functioning and working memory after upper limb vibratory treatment.

With respect to executive functions after treatment, we observed an improvement in the ability to switch attention between two rules or tasks and cognitive flexibility (TMT B). This improvement was not maintained in the follow-up. In contrast to our results, previous studies reported no changes in this function after motor treatment [47,48,49], possibly owing to differences in the type, frequency or duration of the interventions. The abovementioned studies focused on gait and balance treatment, whereas we trained the upper limbs exclusively. In contrast to gait [50,51,52], during upper limb movements, prevalent cerebral activation occurs in the cortical areas (i.e., sensorimotor cortex, supplementary motor area, premotor cortex, inferior frontal gyrus and ipsilateral anterior cingulate) both in neurological patients and healthy controls [53,54]. Motor exercise may increase the production of growth factors and promote gray and white matter changes, especially in the prefrontal region, with a particular effect on executive functions [55,56,57]. On the other hand, during executive tasks, improved performances were associated with an increased activation of cortical regions [58,59,60]. Accordingly, specific motor training with upper limbs, which enhances cortical activation in a different manner than lower limb training, could facilitate an improvement in executive functions (switching attention and cognitive flexibility). However, this hypothesis should be tested by brain imaging studies in the future.

We observed an improvement in visual selective attention (TMT A), psychomotor speed, visual–motor coordination, concentration (DSym) and alertness (ALL_CA). With respect to selective attention, our results are in line with those reported in previous studies [13,61]. To the best of our knowledge, this is the first report on an investigation of attention aspects before and after upper limb motor training (i.e., psychomotor speed, visual–motor coordination, concentration and alertness). Considering the impact of attention on the lives of pwPD, these data could be useful with respect to the choice of rehabilitation path to improve attentional skills during motor activities.

Concerning motor aspects, after treatment, we observed an improvement in global motor functioning (MDS-UPDRS—part III) and a reduction in the disability of the upper limbs (DASH), which was maintained at follow-up. These data are in line with evidence reported in the literature [62,63,64].

In agreement with literature, our data indicate no effects of upper limb motor treatment on global cognitive functioning (MoCA) or working memory (DSB).

Specifically, previous studies indicated no improvements in global cognitive functioning after anaerobic lower limb treatment [22,47], whereas a significant change in the MoCA score has been reported after aerobic programs (e.g., dance or bodyweight exercise treatments) [19,21]. It is possible that different types of exercise (aerobic vs. anaerobic) have different effects on global cognitive functioning. Our intervention involved activities performed with the upper limbs, which can be considered anaerobic training. We hypothesized that although the MoCA adequately considers psychometric properties for a brief assessment of overall cognition in pwPD [40], it is not able to detect cognitively specific changes after an anaerobic program. Therefore, function-specific tests could be used to provide clear indications of cognitive functioning. Accordingly, we evaluated multiple cognitive domains before and after treatment.

Previous studies indicated that a short motor treatment, such as that investigated in the present study, has no effects on working memory [19,49]. Conversely, an improvement in working memory was reported after a 24-month training program [61]. Therefore, memory functions seem to be influenced by the duration of the treatment.

The present study is subject to some limitations. Because individuals were not tested “off” medication, no conclusions can be drawn about the unmedicated state. A sample size with greater gender and age balance could better validate our results. Another limitation is the lack of a control group, as a placebo effect has been reported in several studies involving pwPD [65,66]. Further investigations are needed to compare the effects of different motor rehabilitation approaches with the effects of combined programs (cognitive plus motor) on motor and cognitive abilities. Finally, it will be useful to investigate whether the reported treatment could also be beneficial in pwPD in an advanced stage of the disease who are more compromised. Despite our attempts to maintain the blinding of outcome measure assessors, masking might have been incomplete. Given these limitations, caution is warranted in generalizing the results to the population of pwPD, and additional research is warranted.

In summary, our results seem to indicate that upper limb vibratory rehabilitation promotes an improvement in cognitive and motor skills in pwPD, which can be explained by the complex interplay between the basal ganglia and cortical–cerebellar networks in the modulation of both cognitive–motivational and motor aspects of action [67,68], which have been widely studies in recent years.

The aim of our study is in line with the growing interest in cognitive deficits in pwPD, which frequently impair social function, intensify caregiver burden and increase the costs of disease-related medical care [69]. Accordingly, it is important investigate the influences of motor rehabilitation on cognition. Studies have suggested a significant correlation between functional mobility and global cognitive functioning and attentional ability in pwPD [70], with improvement in cognitive function reported following several kinds of physical intervention for lower limbs [13,71,72]. To the best of our knowledge, this is the first study to investigate the effect of upper limb motor rehabilitation on cognition in pwPD. Further investigation of cognitive–motor relations can improve understanding of the complexity of PD and aid in the development of appropriate rehabilitation programs.

## Figures and Tables

**Figure 1 brainsci-12-01684-f001:**
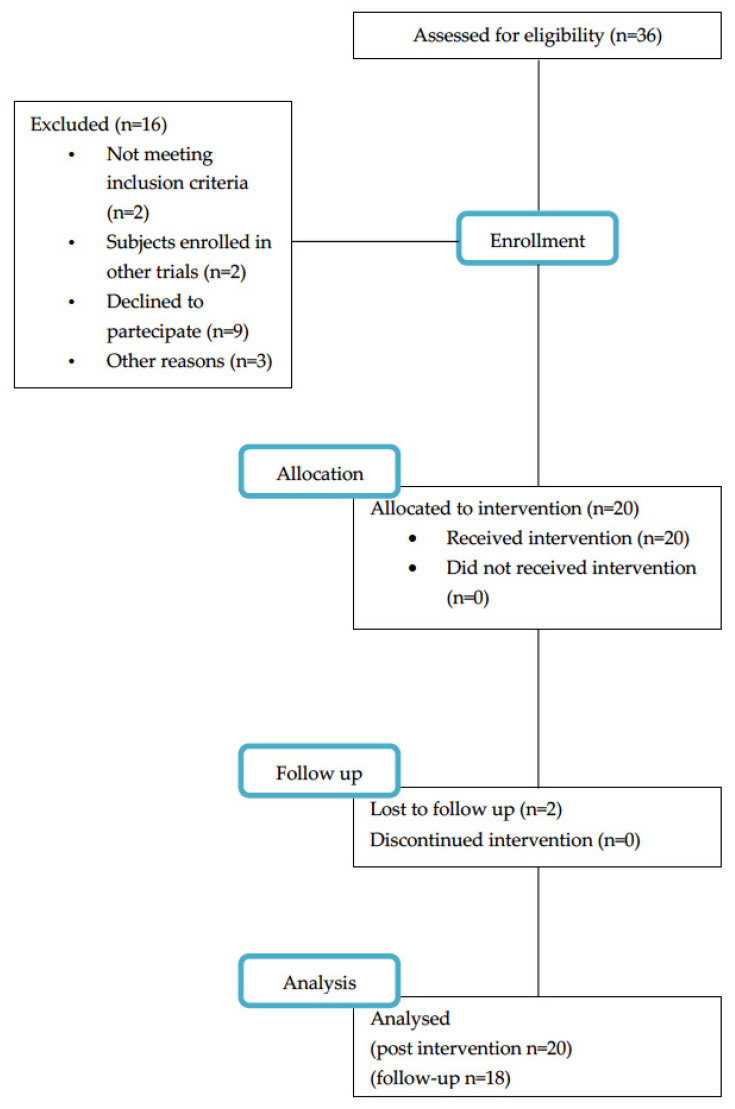
Study flow.

**Table 1 brainsci-12-01684-t001:** Within-group comparisons for outcome measures analyzed using non-parametric tests.

Outcome	Pretreatment	Post Treatment	Follow-Up	Within Group Comparison	
				Post vs. Pretreatment *p*-Value	Follow-Up vs. Pretreatment *p*-Value
TMT A (seconds) mean (SD)	78.47 (40.48)	65,63 (36.12)	86.88 (49.07)	0.011 *	0.309
ALL_SA (mseconds) mean (SD)	429.60 (197.76)	367.44 (135.33)	381.44 (124.94)	0.033 *	0.049 *
ALL_CA (mseconds) mean (SD)	402.55 (164.07)	347.2 (116.42)	350.56 (89.09)	0.009 *	0.079
MDS-UPDRS—part III (0–136) median (IQR)	20 (16.75; 38.25)	18 (13; 33)	18.50 (14.5; 30.75)	0.005 *	0.028 *
DASH (0–150) median (IQR)	47 (36; 51.25)	39 (32; 43.75)	36 (32; 46)	0 *	0.001 *

Abbreviations: IQR = interquantile range; SD = standard deviation; TMT A = Trial Making Test Part A; ALL_SA = Alertness without advertisement; ALL_CA = Alertness with advertisement; MDS- UPDRS—part III = Unified Parkinson’s Disease Rating Scale—part III; DASH = Disability of the Arm, Shoulder and Hand Questionnaire; * = statistically significant (*p* < 0.05).

**Table 2 brainsci-12-01684-t002:** Within-group comparisons for outcome measures analyzed using parametric tests.

Outcome	Pretreatment	Post Treatment	Follow-Up	Repeated Measures ANOVA *p* Value	Pos Hoc Analysis Whitin-Group Comparison	
					Post vs. Pretreatment *p* Value (95% CI)	Follow-Up vs. Pretreatment *p* Value (95% CI)
MoCA (0–30) mean (SD)	22.35 (4.92)	23 (4.69)	23.06 (3.26)	0.468	/	/
TMT B (seconds) mean (SD)	179.63 (81.19)	153.95 (74.96)	208.76 (93.57)	0.045 *	0,011 (7.53; 49.88) *	0,389 (−56.94; 23.41)
DSym (0–93) mean (SD)	21.94 (9.16)	24.47 (8.17)	23.71 (9.47)	0.004 *	0.003 (−3.97; −0.97) *	0,007 (−4.65; −0.87) *
DSB (2–8) mean (SD)	3.80 (1.23)	3.65 (1.08)	3.78 (1.11)	0.611	/	/

Abbreviations: SD = standard deviation; CI = confidence interval; MoCA = Montreal Cognitive Assessment; TMT B = Trial Making Test—part B; Dsym = Digit Symbol; DSB = Digit Span Backward; * = statistically significant (*p* < 0.05).

## Data Availability

Data are contained within the article or Appendix A.

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
