# Peer review of "Effect of Upper Limb Motor Rehabilitation on Cognition in Parkinson’s Disease: An Observational Study"

_brainsci, 2022, doi:10.3390/brainsci12121684_

Round 1
Reviewer 1 Report
This is a novel idea (e.g., effect of upper extremity vibration on cognition), and an important pilot study (although the authors never use the term “pilot study” in their paper). There are limitations including lack of a rationale for the use of a vibratory intervention in PD, lack of a control group (placebo effect), and lack of assessor blinding.
The introduction ought to provide a brief rational for the use of a vibratory training intervention. What is the proposed physiologic mechanism of vibration effect on limb and brain function and ultimately on cognition? For example, other studies have used vibratory socks to improve balance in PD. Or, a vibrating support surface on which the participant stands, in order to improve balance/equilibrium/postural control. This literature in PD could be briefly mentioned. The innovation of the present study and gap in knowledge is use of upper extremity vibration to improve cognition. A major concern is that few of the participant characteristics are reported, such as Hoehn and Yahr stage providing/listing the mean and SD of H&Y, disease duration, co-morbid conditions (one would expect some co-morbid conditions for people at H&Y stage 3 or 4?), ambulatory status (use of assistive devices) of the participants.
Tables are well done and all necessary
All citations are up-to-date
Overall, the paper is very interesting but could benefit from extensive grammatical and synthax editing
Specific comments:
1 Use people first language throughout the manuscript: e.g., instead of referring to participants as “subjects” or “patients” (people are referred to as patients while in hospital only), please refer to participants as “people” or “individuals“ living with Parkinson disease (e.g., iwPD or pwPD).
2 Introduction: line 46-47. edit “about motor aspects…” to “ Motor aspects of PD include problems with gait, postural control and tremor.” Page 2, line 54-55: add the populations in which “clinical and preclinical studies have shown different mechanisms…”. A bit more detail, please. Are these animal studies (rat, mouse)? What model is being used (stroke, traumatic brain injury? Etc). Please add a few details. Line 66 – edit the phrase “during time” to “over time”. Edit “indeed it has been showed…” to “”Indeed is has been shown…”. Line 66-68: edit the last sentence (as it is presently a run-on sentence) into two sentences. Line 77: do not start a sentence with the word “about”. Line 79-81 Edit this sentence to: “Daily upper extremity training Improves short-term memory, attention and executive.” Then follow that sentence with the gap in knowledge, namely, few studies have examined effect of upper-limb training on cognition in PD and the aim of the study was to advance knowledge in…Please also consider to include a central hypothesis. It is important to state an a-priori hypothesis.
3 Materials and methods. State what type of sampling strategy was used. Was it a convenience sample, random sample, or consecutive sample? Page 3, Line 123: “to” ought to be “3” because participants underwent 3 test cycles (baseline T0, post-test T1, and follow-up T2. Line 125: change “cognitive measures are performed” to “cognitive measures were performed”. State whether the test assessors were blinded (e.g., masking of outcome measures assessors), and how the masking was maintained.
4 The intensity of the Armshake vibratory stimulus, frequency of repetitions and number of vibration trials per training session ought to be provided (mean, standard deviation and range).
5 A description of each of the exercises that were administered during the study ought to be added, along with citations.
6 How was the intervention adapted to each participant? (page 3, Line 118). Was the intervention tailored each session, each week, or how? This should be described in enough methodological detail so that other researchers can replicate the study. Please report the average number of oscillations of the device, mean/standard deviation and range duration of the exercises, and rest time.
7 Provide the test-retest correlation coefficients (and literature citations) for use of each primary and secondary outcome measures in people with Hoehn and Yahr stage 2-4 Parkinson disease.
8 Page 4, Line 176. Is LSD the Tukey LSD? If so, please spell out the acronym for “LSD”.Line 177: Please change the term “parsed” to the term “analyzed”.
9 List the a-priori sample size and power for the primary outcome measure.
1 Results: page 5, Line 190. Change the word “resulted” to “the word “were”. Eliminate the copy in parentheses: Line 190-191 “(KS and SW tests…”).
11. Results page 5: Line 196: delete the words “respect” and replace them with the word “compared” and line 197 can be edited from “…and return comparable to baseline at follow-up…” to “…and no change from baseline to follow-up….” Please add the Tukey cut off value.
12. Results: page 6: edit the first sentence of the first and second paragraph under section 3.1.3 to “In regard to…”
13. Results: It is not necessary to repeat that the ANIOVA had 3 levels because authors already provided that explanation earlier.
14. Results: edit the sentence on Line 213-214 to “Table 1 and 2 present the group data and results of the within group analysis.” Add the word “analysis” to the end of that sentence.
15. Please report the following variables for the participant characteristics from the participants chart or health history (if known): ambulatory status (use of assistive devices (– how many used assistive devices, what kind were used); medications used at baseline and at post and follow-up test (– amount and kind); number of years since diagnosis with PD; co-morbid conditions (e.g., diagnosis with depression (or depressive symptomatology, if known), apathy etc.
16. Discussion: The discussion ought to be rewritten. In the first sentence of the first paragraph of the discussion repeat the study aim. The second sentence of the first paragraph of the discussion list the key results 1,2,3, etc. In the second paragraph of the authors ought to discuss, discuss their results in the context of other studies. The discussion would improve in flow and readability if authors use the same sub-headings in the discussion as were used ion the results section. The paragraph on limitations really needs to focus on the methodological weaknesses of the present pilot study: lack of blinded outcome assessors, lack of a control group, most participants were male, older, therefor generalization to females with PD or younger onset is not possible etc.
17. The placebo effect is well known in Parkinson disease and is, in fact, a major issue in non-pharmacological interventions. The discussion should discuss (or at least acknowledge) the possibility of a placebo effect. A placebo effect could explain the entire results because the authors did not include a control group.
Author Response
Thank you for your revision. We revised the manuscript as you suggested. Please see the PDF file for the details.

Reviewer 2 Report
Introduction: The authors say that there is no literature about effects of upper limb training on cognitive and motor skills. It can be helpful to define and discuss some of the currently available exercise programs or routines focusing on upper limbs for the readers.
Methods:
1. How long after the end of the treatment period were the participants followed-up for testing?
2. Did the authors ask the participants about other cognitive training/speech therapy/cognitive activities that participants may have done or engaged in during the course of the study?
3. Was education status of participants collected for this study? If so, that is important to include in the participant demographics.
4. The authors discuss that the participants were asked to maintain a postural stance as erect as possible. From the stance perspective, is it completely an upper limb phenomenon or is a coordination needed between upper and lower extremities to maintain a proper posture and erect stance?
5. Were all participants on dopaminergic medications? Or were participants also with deep brain stimulation?
6. Table 2: Typo for “within group” heading
7. What other physical therapy program did these participants report prior to the beginning of the study? Can the authors speculate about possible cognitive changes in participants who have been receiving physical therapy from before than those received services for the first time in this study?
8. Were the participants tracked for their self-rated effectiveness of medications and motor symptoms during the treatment sessions? I wonder if participants with similar ratings may have exhibited similar performance levels on the motor training program.
Discussion:
1. Depression can affect one’s cognitive functioning. Was the depression status of participants taken into account here? It will be helpful to included whether or not depression status of participants was considered in this study.
2. One of the limitations of the study is that authors used a cognitive screener to determine the possible impact of the 3-week upper limb motor rehabilitation program. Montreal Cognitive Assessment (MoCA) is a cognitive screener and may not be adequate as the only cognitive test measure. With cognition as one of the secondary outcome measures, I wonder if the authors could have considered using additional cognitive assessments like Dementia Rating Scale (DRS-2) or Scales for Cognitive Communicative Ability for Neurorehabilitation (SCAAN). These cognitive assessments are very helpful to obtain a detailed viewpoint about how attention, memory, problem-solving, and visuospatial skills may have been influenced by a physical/motor/cognitive rehabilitation program.
3. One additional limitation is the gender distribution among the participants. Only two were females with PD.
Author Response

(The authors gave the same response as above.)

Round 2
Reviewer 1 Report
Abstract
1. Line 27 and Line 38 and Line 40, uncapitalize the word “Disease”.
2. Line 37: change “findings” to “Results”. Eliminate the word “our”.
Introduction:
1. Line 45 uncapitalize “Disease”, also throughout the paper, (except, of course, in instances where the term Parkinson is used in the context of the UPDRS), UKPD and other terms, etc.
2. Line 49, add specific percentages to the statement that people with PD “often” exhibit cognitive impairment at diagnosis.
3. Line 51, you might want to mention that DBS has been shown to effectively improve cognition long term in people living with idiopathic PD (and add a citation).
4. Line 54: animal and human “models). Plural.
5. Line 68-70 fix the grammatic and synthax errors in this sentence (e.g., “…ones differently….”).
6. Line 76: “regarding”, not “regards”
7. Line 89-90: be more specific: what sensitive areas are activated? Th basal ganglia, the motor cortex? What is meant by the term “sensitive”?
8. Line 90: stimulations” Or do you mean vibratory stimuli? Are there studies? Please cite them and briefly what was found.
Method
1. Line 106: change “people” to “participants”.
2. Line 107: briefly identify what the OSF trial focus is, is it an RCT? What was the aim? Briefly explain.
3. Line 112: change “people” to “participants”.
4. Line 122: explain whether the intervention was administered in a group setting (if so, how many per group on average (range and SD)), or one-on-one.
5. Line 134: change “sit down” to “seated”
6. Line 140: change “beyond” to “In addition to…”
7. Line 143: change “performed” to “administered”
8. Line 146: change “Medical doctor” to “physician” and add what type of physician specialty: rehabilitation physician, neurologist?.
9. Line 147: how was the blinding of the assessors maintained or assured?
10. Line 151: provide the test-retest correlation coefficient for the Trailmaking test in PD.
11. Line 185: change “individual” to “participant”.
12. State that participants were tested in the “ON” state (or in the “OFF” state, if true).
13. In the statistical analysis section, state that results are presented as means and standard deviations (SD). IN the results section eliminate the term “mean” (it is used repeatedly)
Results
1. Line 205: mean education age. Why “age”?
2. Line 217-218: eliminate the sentence: “The remaining outcome measures were 217
performed by Wilcoxon signed ranks test.” Authors already have explained in the statistical analysis section how data were to be analyzed.
3. Line 226: eliminate the phrase: “In regard to outcome measures analyzed with nonparametric tests,”
4. Line 233: Edit to “The one way ANOVA produced a significant time main effect.
5. Line 233-238: what is “DS”? If DS is SD (standard deviation), only report SD values to 2 decimal points, (not 3).
6. Line 235: for the F ratio: why is the η=0,297 provided? (F(2,32)=6,758; P=0,004; η=0,297).
7. Table 1: why are the results for TMT A bolded I this table?
Discussion
1. The second sentence of the first paragraph of the discussion should list the key results: 1,2,3.
2. Line 272: eliminate “About to attention abilities after treatment”
3. Use headings in the discussion: e.g., working memory, limitations etc. Use the same headings in the discussion as authors used in the results section.
4. Line 305: change “another important limit….” To “another limitation…” Authors should add that because of these limitations caution is warranted in generalizing the results to the population of pwPD. More research is warranted. Another limitation is that upper limb “freezing” and/or apraxia and/or dexterity and/or “proprioception” were not assessed, or were they?
5. Line 318: change “literature evidenced….” To “studies suggest….”.
File 1: the exercise forms. Is this tip-sheet copywrited? Please also provide a citation and web link to the exercise “sheet
